# Notes on the Present and Future Research on World Literary Journalisms

**John S. Bak** 

Department of English, Université de Lorraine, 54000 Nancy, France; john.bak@univ-lorraine.fr

**Abstract:** This review article offers a glimpse into the problems and promises of current research on world literary journalism. It discusses the rise and spread of press cultures via colonialism, the contentious nature and taxonomy of the *fact*—its subjectivism, accessibility, and veracity—within an inconsistent global press, and how the porous divide between fiction and nonfiction genres is affecting the production and consumption of literary journalism around the world. The article concludes by offering nine areas of research (from canon-building and historiographies to digital news platforms and gendered media) still under-represented in international and transnational literary journalism studies.

**Keywords:** world literary journalism; reportage literature; transnational research

## 1. Introduction

Literary journalism is as timeless as it is ubiquitous. From the rocks of Kakadu to the cave walls of Lascaux, art has documented the sacred and the profane of peoples' lives for millennia. No one period can lay claim to its invention, no one culture its ownership.[1]

Like all documentary art, literary journalism spouts from the human necessity to leave behind an accurate record of its passage on this planet, and, along the way, the genre has taken on many shapes and styles. Historically, a narrative rending of fact and truth—from travelogues to jeremiads to almanacs and from political pamphlets to missionary dairies to oral readings of the popular press—was seen as a way to empower the body politic, or genuflect to the crown. Certain regimes demonized the form, fearing that it would enlighten the ignorant or misinformed, or expose the corrupt and maleficent. Some nations never even saw literary journalism as being independent of the realist fiction or the popular press that was already firmly entrenched in their belles-lettres. How one nation sees and understands literary journalism today is thus inextricably tied to that nation's earlier print culture and political development. And given that facticity and aesthetics, the epistemic pillars on which literary journalism firmly rests, are themselves open to historical scrutiny and cultural interpretation, it is perhaps prudent to talk about literary *journalisms* in their plurality. While the term appears in this article in its singular form, readers should understand and celebrate it in this collectivist spirit.

Undeniably, there have been watershed moments in the genre's extensive history, from French *feuilletons* and undercover journalism in Australia in the nineteenth century to New Journalism and the Latin American Boom in the twentieth. For this reason, scholars have accepted the paradigm that literary journalism was born in the nineteenth century, matured in the twentieth, and is now going digital in the twenty-first. Each century, though, is the product of a long and complex process of bilateral transnational influences and intermedial hybridizations, the ineluctable results—and, at times, denouncements—of colonialist expansionism, rampant urbanization, and widespread diasporas or displacements of peoples and cultures. The many parameters involved, each a morass in its own right in terms of cultural mediation, problematize any historiography of one country's literary journalistic creation and development. In this sense, literary journalism is like religion, culturally

innate but discriminately migratory, whose many mutations were catalyzed by periods of great human confluence, forced or otherwise.

Religious traces themselves are even palpable in the genre's development over the centuries around the world. British Protestantism, for instance, honed early North American plain style and inculcated its current fact-checking rigor, just as Iberian Catholicism fused with indigenous Latin American myths and symbols to create the firebrand *crónica* and the dishy *crônica*, forms that first developed from—and later influenced—Spanish *periodismo narrativo* and Portuguese *Jornalismo Literário*. Just as the world eschewed religious singularity, it equally resisted the adoption of one form of documentary art, and often for the same reasons: the circumstances under which the confluence was initiated (predatory colonialism) and the means by which interculturality was enforced or suppressed (cultural hegemony). Like the many postcolonial literatures left in the wake of devolution, world literary journalism has had to adopt, adapt, and adept, but never in the same manner or to the same degree.[2] In short, it is not just the name *literary journalism* which has been open to debate over the years; it is also its fundamental constitution, from one culture to another, from one continent to another. As a result, we find today as many vibrant pockets of literary journalistic activity throughout the world as there are telling dormancies.

Over a decade ago, *Literary Journalism across the Globe*, the first book to address the history and praxes of itinerant and transgressive modes of the genre throughout the world, informed its readers that the U.S.'s New Journalism, despite its extensive world reach and influence (Alexander and Isager 2018), was neither the source of world literary journalism nor its sole champion. New Journalism was surely the most celebrated proponent, which made the world identify literary journalism for years as an American form. Years of international research since that book's publication, however—whether delivered orally during one of the annual congresses of or outlier panels for the International Association for Literary Journalism Studies (IALJS) or in print in any number of journals now solely (*Literary Journalism Studies*) or partly (*Textos Hybrides*, *Brazilian Journalism Research*, *Journalism*, *Prose Studies*) dedicated to publishing research on international literary journalism—have made it clear that literary journalism is not only a global phenomenon, but has been for centuries.

As our knowledge of the form's many histories take shape, and as our national canons expand through discoveries or recoveries of prior texts and their forgotten, censored, or expunged authors, it is time once again to take stock of the advances made in the field, and to look more inclusively as to where the world's literary journalisms came from, what fuels their present renaissance, and where they will likely proceed in the decades ahead.

## 2. Black Coffee, White Milk, or Fifty Shades of Truth

Given what has just been described, it is unnecessary, perhaps even imprudent, to establish a definition of literary journalism. The IALJS's precept, "'journalism as literature' rather than 'journalism about literature,'"[3] will thus serve as this article's touchstone in all matters taxonomic concerning the genre's international manifestations. Be it the New Journalism, the Latin American *crónica*, or the European *reportage*, they all incarnate pretty much the same spirit and motivations behind producing narrative nonfiction, often to redress the ills visited upon the presumed residuum of a nation. Most readers drawn to this article will already have a working knowledge of what literary journalism is or means to them, what it represents today for the future of print and digital media, and what its potential limits and drawbacks are. What these informed readers may not be familiar with, however, are the various nuances that distinguish a country's, a region's, or even a language's traditions of literary journalistic reporting and writing. For those who do not possess such prior knowledge of the genre, nor of its current global praxes, this article will present nine of what I consider to be the most pressing issues facing literary journalism in the world today, and by extension literary journalism studies. It is, therefore, opportune to look briefly at the various issues and debates that inform the arguments of the many chapters that follow.

You can learn a lot about a country, or a culture, from the way its people collectively take their coffee—or their tea. Black, or white with milk, or somewhere in between. A true Brit, for instance, will scold you for putting the milk in first—that takes liberties and makes unwelcome presumptions. When Italians order *un caffè, per favore*, they know it will be an espresso by default. For the Spaniard, equal parts of coffee *con leche* kickstart a morning routine, while a simple black *café solo* is reserved for the afternoon bar or *cantina*. For the Turk or Greek (*sade kahve* or *sketos* . . . careful here), the national java is a muddy coffee where the finely ground lees gather in the bottom of a demitasse accompanied by a glass of water (to wash down any sediment that finds its way into a sip). A diluted black coffee from drip, instant, or (less often perhaps) percolate is the standard North Americans brew (referred to as *americano* in cafés around the world), traditionally drunk anytime during the day at home or served in bottomless cups at greasy-spoon diners (though today's commercial espresso machines or super-sized to-go concoctions from one the many multinational chains are bucking this tradition). Brazilians like dripping their coffee into large thermos pots that can be slowly drained all morning, and Peruvians serve up an intense, instant coffee syrup in a pot alongside a mug of hot water for diluting purposes.

On the other end of the spectrum, there are the pure milk drinkers, be it from a cow or a goat or, as parts of the world turn vegan, processed from oats, almonds, or soy. Perhaps a cup of steamed milk, often with a drizzle of honey, will be their drink of choice.

Others, of course, mix them. There are those looking for strong black coffee whose bitterness is cut by the milk's lactose, and those seeking a dairy drink whose sweetness is tempered by the coffee's roasted lactones. Because the proportions of one to the other are almost never equal, nations and languages have invented multiple terms, as the Iñupiat have with snow, for the resulting coffee–milk blend. Each denotes with precision the various degrees, shades really, of black coffee to white milk. And again, you can tell a lot about a nation by the nomenclature it adopts, further evidence of the linguistic determinism or relativity first described by Edward Sapir nearly a century ago and taken up by his student Benjamin Lee Whorf.[4] Heartland U.S.'s black coffee drinkers may ask for half-and-half, cream, or milk, while Bostonians know that a "regular" means "with cream and sugar". The Aussies' (or the Kiwis') flat white, adopted en masse by the British, is an espresso with a small amount of steamed milk and a thin layer of microfoam. The French have their *espresso*, their *noisette* (a drop of milk in an *espresso*), their *café crème* (a bit more steamed milk), and their *café au lait* (often served in equal portions in the morning in a small bowl; like the Italians and the Spanish, the French *do not* drink milk coffee in the afternoon or evening).

The kings of coffee terminology are, of course, the Italians. Like tennis vocabulary for the English, golf terms for the Scottish, and French culinary lingo, coffee's lexemes are in Italian. Because an espresso is Italy's standard, a spectrum of words for portions of black coffee to white milk had to be established to accommodate those who preferred a different cup of joe. Though the terms have been known around the world for decades, Nespresso and Starbucks have certainly consolidated the coffee myth of Italy and made Italian the lingua franca of coffee culture: *corto*, *stretto*, *macchiato*, and *caffè macchiato* on the dark end, and *cappuccino*, *piccolo*, *latte macchiato*, and *caffè latte* on the lighter end. So common are these words today that they no longer need appear on sandwich boards or in menus in italics.[5]

The astute reader will deduce where this is headed. Substitute journalistic "objectivity" or "fact" for "black coffee" and narrative "subjectivity" or "fiction" for "white milk" (or vice versa) and a similar spectrum emerges, one that equally informs us about a nation's or a language's taste in literary journalism. The terms used around the world are many and are, in principal, synonyms of each other; but just as with the language of coffee, they are not identical and exist instead along a continuum between objectivity and subjectivity and between fact and fiction: literary journalism, narrative journalism, New Journalism, *crónica*, reportage, *Jornalismo Literário*, *periodismo narrativo*, *journalisme littéraire*, *testimonio*, очерк (sketch), *literarische Reportage*, creative nonfiction, الصحافة الأدبية (literary press), etc. Many are direct translations of the English "literary journalism", but that is a recent phenomenon,

one still open to debate in certain countries more than a little proud to suggest that their rich traditions of narrative journalism followed the North American model.

In fact, we speak of literary *journalisms* in the plural because some nations (such as the U.S.) take their literary journalism "black"—bitter in its unwavering facticity—whereas others (like Australia) prefer a "flat white" of creative nonfiction. Latin American nations, once under the control of censoring juntas, turned to an arabica–robusta blend that most readers understood as objectivity masquerading as magical realism's subjective idealism,[6] a hybrid not dissimilar from Germany's *Neue Sachlichkeit* (New Objectivity). As Christopher Warnes writes, "A magical idealist is one who participates in the project of apprehending truth not through correspondence with external reality, but by undoing the antinomies between language and the world and between subject and object".[7] Prior to the consolidation of literary journalism studies as an international discipline, various countries simply called it "journalism" or "realist fiction". As with the ordering of a simple coffee before the current java trend, there was little need for complex taxonomy. In short, one literary journalism, today and historically speaking, is not like another, and however you "take" yours in the morning, afternoon, or evening, may or may not be how another person takes his or hers. Neither is correct; purists are not the rule.

The problem in identifying a singular, transcendental literary journalism, then, lies in the problem of first isolating its etymologies and epistemologies. Unlike coffee and milk, objectivity and subjectivity, like fact and fiction, are not binaries, nor mutually exclusive. Ordering a black coffee and being served instead a glass of warm milk would prompt one to send it back, and the barista would apologize accordingly. Nowhere on planet coffee is one a substitute for the other. Such is not the case with objectivity and subjectivity, however—or with fact and fiction.

Problems can be located at the micro and macro levels, beginning with terminologies and definitions and expanding toward epistemes and praxes. Take, for example, the words *fiction*, *fictionality*, and *fictivity*. The "narratological traditions" of English, French, and German, as Monika Fludernik and Marie-Laure Ryan write in the introduction to *Narrative Factuality* (Fludernik and Ryan 2020), "have used these terms in quite a distinct and to some extent incompatible manner".[8] Since fiction has had a long tradition in German-language literary journalism as an "an effective and well-established instrument to increase the authenticity of journalistic reporting", Tobias Eberwein posits that the German distinction between *fiktional* and *fiktiv* can also be applied to *faktual* and *faktiv*.[9] Fludernik and Ryan continue:

> Having neatly separated the fictional and the fictive (fictionality and fictivity), German terminology uses *Fiktion* for the abstract concept of fictionality (in its English meaning), and *never* for the genre of fictional texts or novels (English: *fiction*). By way of analogy with the *fictivity/fictionality* doublet, German scholars have recently proposed . . . introducing the same type of distinction for the realm of factuality, thus suggesting an opposition between *factive* (*faktisch*) persons, places or events and *factual* (*faktual*) texts, statements and discourses. This usage is new in German language narratology but meets wide acceptance as an innovation; however, French and English narratologists are largely opposed to the German distinction of *fictive* vs. *fictional* (which is new to them) and are therefore even less sanguine about extending it to the realm of the factual.[10]

If, despite its seemingly sound foundation, the word *fact* is as nebulous as, say, *narrative*, how are we ever to agree upon terms such as *truth* or *objectivity*, which are wholly dependent upon it? This is not a postmodern hair-splitting exercise separating sign from signifier or semiotic gymnastics. It is a real, tangible distinction between languages and cultures that has ramifications from the field to the newsroom. It is not the goal here to debate this distinction per linguistic determinism per the Sapir–Whorf hypothesis but rather to point out its fundamental application to literary journalism studies. As Fludernik and Ryan rhetorically ask, ". . . do factual and fictional narrative constitute positively defined

opposites, comparable to black and white, or is fictional narrative only one member of a broader field of nonfactual genres and modes of narration that contains many shades of grey?"[11]

Similar problems surround the word *truth*. On the macro level, we often associate a truth, like fiction with fact, as the mirror opposite of a lie, but philosophy, sociology, anthropology, and pretty much every *ology* have shown us that this is rarely the case. Fiction is something made up—for some a lie, for others an untruth—but nonfiction is not entirely a truth, and a lie can become the truth if there are enough people collectively advancing its narrative. To speak an untruth deliberately is to lie, but an unintentional lie is not ipso facto a fiction. Burdened by its negative prefix, nonfiction has been historically set up as fiction's other and thus inclusive of everything not deemed (dare I say valued) as fiction. We do not, for instance, refer to reportages as faction and novels as nonfaction. But fiction could have been truthful at one point (epic history, myths, realism), and even serve nonfiction later (Maguire 2015; Clingman 2012), just as nonfiction may have been a lie (history, epistles, memoirs, autobiography). How a fact was rendered truthful, then, is as important as how it is later represented in a narrative. For example, if an eyewitness believes what he or she saw and recounts it as truth but which was, historically speaking, a wrong or an incorrect assessment or a culturally informed or biased take on the events that transpired, the journalist documents that statement as factual when it potentially was not.

And when *truth* is set up alongside *fact*, problems become even more pronounced. There are boldface lies, which intrinsically oppose the truth, just as there are *untruths*, intentional or otherwise that belie a lie. "Today is Monday" may be true for the Inupiat living on Little Diomedes (U.S.), should it actually be Monday, but for someone on Big Diomedes (Остров Ратманова in Russian, or Ratmanov Island)—separated by only 2.4 miles (3.8 km)—it is likely already Tuesday. So, the statement is both a truth and an untruth simultaneously. It is not a lie because a lie is an intentional falsity and at no point in its utterance is it true or untrue.

French narratologist Gérard Genette's "Fictional Narrative, Factual Narrative" (Genette 1990) exposes the slippery "interaction between the fictional and factual domains of narrative" and shows how "heterodiegetic fictional narrative is in large part a mimesis of factual forms, such as history, news articles, and reporting" but that "this is a simulation whose marks of fictionality are optional and can very well be done without".[12] "[R]eciprocally", he adds,

> "fictionalization" . . . [has] in recent years become widespread in certain forms of factual narrative, such as reporting or investigative journalism (what in the United States is called the "New Journalism"), and related genres such as the "nonfiction novel".[13]

Because of their nonexclusionary natures, fiction and nonfiction, Genette concludes, fail to sustain a binary system that literary journalism studies so preciously requires:

> Such reciprocal exchanges tend to attenuate considerably our hypothesis of an a priori difference between the fictional and non-fictional narrative systems. If one limited oneself to pure forms, free from contamination, which no doubt are only to be found in the poetician's test tube, the clearest differences would seem essentially to involve those aspects of mode most closely connected to the opposition between the relative, indirect, and partial knowledge of the historian and the elastic omniscience enjoyed, by definition, by someone who invents what he narrates. If one took into consideration actual practice, one would have to admit that there exists neither pure fiction nor history so rigorous as to abstain from all "plotting" and all novelistic devices whatsoever, and therefore that the two domains are neither so far apart nor so homogeneous as they might appear.[14]

Words and concepts like *fact* and *fiction* and *truth* and *lie/untruth*, especially in their international currency, are a blur at best, which is why the corpus of ontological and academic studies on them is so incredibly vast.

If we look at the problem from another, more tangible angle, perhaps we can understand the complexities in a different light. To say something is *green*, for instance, may be a truth or an untruth depending upon the perceiver and what the eye distinguishes as green and what that perceiver's culture and language understand blue to be in the color spectrum. Their culture may have few or multiple names for green because there are so many hues present in the color spectrum in that culture's accepted palette. Consider the color green to the Inupiaq people mentioned earlier, who see (or, rather, saw, given the change in climate) the blossom of spring less frequently and intensely, versus their notion of white, or winter ice and snow, which is (or was) much more prevalent in their surroundings. The same could be said about those living in lush green settings where winter dare not venture. Color nomenclature is culturally based, Brent Berlin and Paul Kay famously showed us (Berlin and Kay 1969),[15] with Walter Ong adding in *Orality and Literacy* (2002) that geography plays an important role as well,[16] something cultural anthropologist Franz Boas (Boas 1911) had posited a century earlier.[17]

Unlike fact or truth, which are epistemic notions, that is, acquired through education or indoctrination by families, schools, and religions, the color green is first a somatic and then an ontological concern, that is, perceived physically as a reality and then processed linguistically by a cultural phenomenon. Like the color green, then, a fact (and its sibling, truth) has no polar opposite per se, but instead several hues that depend on white or black for nuance. A fact is denoted as much by *what it is not* as by what it is. And in literary journalism, a form of writing based mainly, essentially, or exclusively on the factual (depending on the country) can create problems or richness, depending on how one perceives it (or wants to perceive it).

The nine avenues for future research proposed below take up this debate, directly or indirectly, in their exploration of their country's literary journalism. Their many antipodal arguments—and, at times, basic principles of literary journalism's nonfiction label—will lead some readers to ponder the implications and differences from their own ideologies of what the genre is, or should be, whereas others might reject the variants outright. This review article is not after a McDonaldization of the genre (or Starbuckian, given the earlier extended metaphor), where literary journalism tastes the same no matter what the country. Quite the contrary, it wishes to celebrate literary journalism's domestic flavors and multivocal agendas. Polish *reportaż* is and is not like the Chilean *crónica*, and that is seen as a healthy sign of the genre's global outreach more than a challenge to its many voices, strains, and appearances. Both, for instance, value editorializing through image as a response to state-controlled censorship, but Polish literary journalism today is found more in book format than in the longform (or even shortform) magazine stories that make up the majority of Latin American *crónicas*. Literary journalism cannot be forced to wear the same shoe, *chaussure*, or *cabot*, and that is perfectly fine, and if one country has more terms for it, like Italy with coffee, it is more a sign of that country or language's rich tradition of the form that needs differentiation, such as the Latin American *periodismo narrativo*, *crónica*, *testimonio*, *perfil*, etc., where facts and information can share not only the same column space with interpretation but also the same story.

### 2.1. Historical Antecedents and Influences

Trying to locate who started literary journalism and then trace its movements from one nation to another is a fool's errand, akin to isolating the first person to wax poetic about the clouds or the sun. Simply put, literary journalism, in one form or another, was everywhere. Traces of it are found in William Bradford's *Of Plimouth* Plantation (1651), just as they are in Samuel Pepys's Diary (1660–69/1825) or James Cook's logs aboard the H.M.S. Endeavor (1768–70) or even the Spanish and Portuguese conquistadors of the fifteenth and sixteenth centuries whose works are known collectively as the *Crónicas de Indias*. Its development as a form, being close to what we identify today as literary journalism, is certainly more identifiable in colonizing countries, whose print cultures developed faster and extended much wider through empire building. Which came first, the popular press or

the *vox populi*, remains an enigma, though. The abolition of the stamp tax and technological innovation (e.g., Hoe's rotary press, teletype, the transatlantic cable), accompanied by lower production costs, growing educational levels, and better living conditions, opened up new audiences for newspapers across Europe, which in turn affected the print cultures of the colonies, especially the Spanish viceroy nations. But what role did literary journalism play in that exchange? It was different for every nation, of course, but in the industrialized nations of the nineteenth century, identifying an "inventor" of the form is fruitless. They all invented it—and in near simultaneity.

There have been a significant number of books and articles on the origins of American and English literary journalism (e.g., Weber 1980; Connery 1990, 1992; Frus 1994; Applegate 1996; Hartsock 2000; Sims 2007; Keeble and Wheeler 2007; Bak and Reynolds 2011; Bak and Reynolds 2022; Keeble and Tulloch 2012, 2014; and Underwood 2013, 2019). To a much lesser extent, though no less important, studies on other nations' equivalents to literary journalism have also been undertaken: Lima (1993), Pena (2006), de Castro (2010), Passos (2014), and Martinez (2016) for Brazil; Merljak Zdovc (2008) for Slovenia; Boucharenc (2004), Thérenty and Vaillant (2004), Thérenty (2007), and Cachin et al. (2007) for France; Meuret (2012, 2016) and Aron (2012) for Francophone Belgium; Pinson (2016) for Francophone Canada; Chillón (1999), Parratt (2003), and Cuartero Cuartero Naranjo (2014) for Spain; Twidle (2012, 2019) for South Africa; Laughlin (2002) for China; Wiktorowska (2018) and Frukacz (2019) for Poland: Soares (2011), Domingues and Trindade (2014), and Coutinho (2017) for Portugal; Poblete Alday (2014) for Chile; Rotker (2005), Calvi (2010, 2019), Mahieux (2011), Aguilar Guzmán (2019), and Chávez Díaz (2021) in collective Latin America. Several shorter pieces have also appeared regularly in the IALJS newsletter, *Literary Journalism*.

Many of these histories and narratives were written in Spanish, French, or Portuguese and remain untranslated, thus being frequently untouched by readers who do not speak the language. This research avenue can not only render some of this historical research more accessible but extend it to other countries that do not have obvious Anglo-American media connections. By exploring how various avatars of current literary journalism took root and developed in various countries, we can learn more about the nature of a proto-literary journalism and its evolution to the genre recognized today.

### 2.2. Literary Journalistic Methodologies

This second research avenue takes stock of the trajectory of existing theories behind and methodological applications of literary journalism theory. As with the scholarship on literary journalism's historical development, there is no shortage of secondary literature on literary journalism's ties, from literary naturalism to narratology. Theories and theoretical research by Sims (1984, 1990, 2007), Lounsberry (1990), Connery (1992), Eason (1990), and Hartsock (2016), among others in the U.S., have been complemented in all parts of the world: Aare (2016) in Sweden; Giles and Roberts (2014) in Australia; Lima (1993), de Castro (2010), Borges (2013), and Martinez (2016) in Brazil; and Eberwein (2013) in Germany and Austria. Each has widened the parameters of an already-vibrant discussion to include other developing theories from around the world, such as Lombroso's positivist theory of criminology where political upheaval and social stratification have been historically more pronounced. A recent issue of the *Brazilian Journalism Research* dedicated to "Literary Journalism as a Discipline" (Bak and Martinez 2018) also contains articles that look into the influence literary journalism is now having on anthropological studies in Latin America. Once the borrower, literary journalism is slowly becoming the lender of theories and praxes, adding to its global currency.

What this special issue on "Literary Journalism as a Discipline" also revealed is that literary journalism is different in Brazil from, say, Germany, in part because the various ways journalists themselves were (and are still) educated and how editorial cultures in newspapers and magazines developed over time. Putting different media histories and cultural referents aside (major elements are already at work in differentiating one nation's

literary journalism from another's, as the histories in Part One have detailed), certain nations have specific universities centered specifically around the teaching of journalism (in France, the *École supérieure de journalisme de Lille*, for instance), whereas others learn the basic methods of reporting and newsroom editing in Communications and Media programs and diplomas, while others still must learn reporting on the job. Since not all literary journalists come with the same foundational education, it stands to reason that their brand of reportage writing will be different as well. While the majority of journalists and editors around the world are familiar with the pyramid formula in writing news stories, what fills the rest of the paper, or online media platform, can vary widely. And the result is that literary journalism, which often finds its home in the marginal sections of the press (for example, in the *feuilletons* that spread from France to other European papers to, finally, the colonial presses around the world), also significantly differs from one country to the next. This avenue could explore to what extent literary journalism extends beyond the traditional aesthetics and methodologies of literature and journalism and even establishes its own theoretical protocols.

### 2.3. War and Conflict

For as long as there have been wars, there has been war reporting. The only thing humankind seems to value more than the taking of life is the recording of that death in ink. From Mesolithic to Neolithic cave drawings at Bhimbetka (India) and Jabel Acacus (Libya) to the Attic histories and epics of Herodotus, Thucydides, and Homer and from Elizabethan tragedies to cult television series like *Generation Kill*, no media, ancient or modern has escaped the theme of man's inhumanity to man, nor has the public's thirst for blood abated with time. For better or for worse, war reporting has remained a rich cultural heritage that touches not only those individual cultures or states that have borne the scars of war on its people or its landscapes, but also the collective memory of what it means to be human—or inhuman.

War and conflict have thus proven to be fertile ground for works of literary journalism around the world: Winston Churchill on the Boer War; John Reed on the Russian Revolution; Martha Gellhorn, Ernest Hemingway, Andrée Viollis, George Orwell, and Langston Hughes on the Spanish Civil War; John Hersey and others on World War II; Mao Dun and Huang Gang on the Second Sino–Japanese War; Michael Herr, Antônio Callado, and Oriana Fallaci from both sides of the front on Vietnam; and Alain Lallemand, Anne Nivat, and Sebastian Junger on Afghanistan.

The first casualty of war is the truth, or so goes the famous quip attributed to a number of different sources, from Aeschylus to the U.S. Senator Hiram Johnson (Knightley 2003). For every literary journalist writing about war, there is a literary journalist scholar writing about their coverage. As with the form's history, the list of scholarship devoted to the subcategory of literary war journalism is too long to cite. Several recent collections published within the ReportAGES series of the Université de Lorraine, where the above quote comes from, have attempted to document literary war journalism from an international and interdisciplinary perspective: World War I (2016), colonial and postcolonial Africa (2018), and Latin America (2019) and Spain (2019). This avenue could give depth and breadth to the great literary war journalism of countries from various corners of the world not yet represented in the secondary literature and discover how literary journalism has changed, or not, depending upon the war and the country reporting it.

### 2.4. Immigration and the Border

Every country has its borders—artificial, intellectual, cultural, geographical—and the meaning of those demarcations nearly always differs when studied locally and globally. While literary journalism studies has often examined the genre's conceptual borders (Bak 2021), those between objectivity and subjectivity or between fact and fiction, many literary journalists themselves deal with real border crossings. Ted Conover's *Coyotes* (1987) and Óscar Martinez's *Los migrantes que no importan* (*The Beast* 2010) track migrants'

journeys tBahrough Mexico to the U.S. border, while a disguised Wolfgang Bauer's *Überdas Meer* (Bauer 2014) accompanies fleeing Syrians from their hiding spots in Egypt on board a refugee boat to Europe. Another example is the Kurdish-Iranian journalist Behrouz Boochani, who fled Iran and its oppressive regime in 2013 but was detained in a refugee center on Manus Island in Australia before being transferred to the Papua New Guinea capital of Port Moresby. His reportage book *No Friend but the Mountains*, tapped out over several years on a cellphone in its original Farsi and sent via WhatsApp and later translated into English, was recently awarded the prestigious Victorian Prize for Literature in Australia and a lucrative movie contract. The world canon of literary journalistic works on borders and border crossings is extensive.

Crossing physical borders is thus a frequent occurrence in literary journalism's praxis. As such, another transdisciplinary tool, border studies, can be called upon to help examine the effects of these crossings, both on the subjects as well as on the writer and his or her readers. Border studies is meant "to chronicle and understand how borders, and border cultures, societies, polities and economies, are not only changing due to major transformations in the global political economy, but also how borders often play key roles in these changes".[18] In their introduction to *A Companion to Border Studies* (2012), Wilson and Donnan provide a detailed summation of the field's goals and methodologies:

> Border studies have become significant themselves because scholars and policy-makers alike have recognized that most things that are important to the changing conditions of national and international political economy take place in borderlands—as they do in like measure almost everywhere else in each of our national states—but some of these things, for instance those related to migration, commerce, smuggling and security, may be found in borderlands in sharper relief. And some things of national importance can be most often and best found in borderlands.[19]

In terms of direct application to literary journalism studies, border studies can help scholars explore not only the socio-political reasonings behind a given country's print culture that makes it either conducive or hostile to genre-fluid textual border identities, but also, in a comparativist manner, the noncontingent notions of literary journalism or reportage literature that lie behind the borders of neighboring countries. Why, for instance, given their close geographical proximity, is French *journalisme narratif* (Vanoost 2012) so different from German *literarische Reportage* (Eberwein 2013), or, for that matter, the German form from Polish *reportaż* (Saignes and Demanze 2021)? While scholars of literary journalism have often focused on the variants in nations' print cultures between fiction and nonfiction, perhaps another way of looking at the problem would be through the various shared "borders" that conjoin neighboring nations. As Wilson and Donnan add,

> Once principally the focus of geography, the study of territorial, geophysical, political and cultural borders today has become a primary, abiding and growing interest across the scholarly disciplines, and is related to changing scholarly approaches to such key research subjects and objects as the state, nation, sovereignty, citizenship, migration and the overarching forces and practices of globalization. All of these approaches to borders and frontiers have been complicated by various attempts to understand and express identities, an effort often related to the investigation of hybridity . . .[20]

Therefore, the "intersection of the metaphorical negotiations of borderlands of personal and group identity (in what has come to be known as 'border theory') with the geopolitical realization of international, state and other borders of polity, power, territory and sovereignty ('border studies'). . ." could be applied to the brackish frontier dividing fact from fiction in nonfiction studies, wherein literary journalism resides.[21]

This fourth avenue could explore the many parameters of borders within literary journalism and its studies, around the topic of migration per media institutions and societal change from the mid-nineteenth century until today; how dominant tropes used to make the phenomena of immigration compatible with media logic(s) and public discourse can

be identified; what journalistic strategies of immersion are discernible in contemporary and historical reportage on immigration; and how new media technologies influence the ways in which literary journalists refine their practice. The literary journalistic treatment of borders and border crossings at several litigious sites around the world remains to be explored in depth, as well as how literary journalism has responded to these different conceptions of borders and their frequent crossings. One can find greater expression of these issues in the online magazine *Words Without Borders* (2003), which publishes, among other things, English translations of longform nonfiction among from around the world.

### 2.5. Female Literary Journalists around the World

In her 2012 keynote presentation at the IALJS congress in Toronto, Nancy L. Roberts brazenly asked, "Do women write literary journalism?" Her answer: yes, but it depends on where you look.[22] The newsroom and the foreign correspondents have traditionally been the domains of men, but women, as early as the fin de siècle, have challenged those gender barriers, broken glass ceilings, and proved that literary journalism is, or should be, gender-neutral. As Karen Roggenkamp writes specifically of the U.S. newsroom,

> Coinciding with the increasing employment of women at major dailies, a robust number of newspaper fictions chronicled the experiences of these professionals who were fighting their way into a male-dominated workplace. Just thirty-five women self-identified as editors or reporters in 1870, a number that grew to 288 in 1880, then exploded to 888 in 1890 and 2193 in 1900—significant growth, though still a small proportion of that year's journalist class, which totaled 30,098. Female reporters lucky enough to secure a desk in the city room faced condescension, opposition, and sometimes open hostility from their male counterparts.[23]

But the story of women reporters and literary journalists is linked the story of gender parity, and any discussion of female literary journalists in the U.S. will be vastly different from that of their counterparts in Latin America, where the struggle for equality was longer and at times more virulent, and likewise, this will differ from discourse in places in the world where the struggle still continues to this day.

Yet women do perceive and write differently than men, and, as Anne Nivat has claimed in *Les brouillards de la guerre* (2011), in certain parts of the world, such as fundamental Islamic nations, a female journalist has an advantage in that behind a burka she can blend in more easily and even penetrate domestic settings where foreign men, especially those in army fatigues, are forbidden. The story of female literary journalists is rich because it also implies the story of gender equality, from the country where the woman is a journalist to the country where she becomes the journalist's subject.

*Literary Journalism Studies* (spring 2015) dedicated a special issue to women and literary journalism, and more and more scholarship is giving rightful due to the female literary journalists who have braved the macho newsrooms or war fronts and trenches (Whitt 2008; McLoughlin 2014; Roberts 2015; Purkis 2016; and Meuret 2015). This fifth avenue of research could build on that initial, groundbreaking research to include not only more essays on under-recognized female literary journalists (e.g., Olive Schreiner, Noni Jabavu, and Bessie Head in South Africa; Oriana Fallaci in Italy; Carmen de Burgos in Spain; Sylvia de Arruda Botelho Bittencourt in Brazil; Alma Guillermoprieto in Mexico; Leila Guerriero in Argentina; Margaret Fuller, Charlotte Perkins Gilman, Rebecca Harding Davis, Willa Cather, Elizabeth G. Jordan, Eva Anne Maddan, Nellie Bly, Anzia Yezierska, Ida B. Wells, and Katherine Boo in the U.S.; Samar Yazbek and Atef Abu Saif is Syria and Palestine; Catherine Hay Thomson in Australia—just to name a few), but also to invite female literary journalism scholars to debate whether or not there can be such a thing as a gender specific literary journalism that is identifiably female (an *écriture feminine* of literary theory), and, if there is indeed such a category, whether or not it is a worthwhile segregation to have.

### 2.6. Censorship and Politics

In his treatise against censorship, *Areopagitica* (Milton 1977), John Milton writes, "who kills a man kills a reasonable creature, God's image; but he who destroys a good book, kills reason itself, kills the image of God, as it were, in the eye". Many a liberal have rallied against demagoguery with such thinking, just as many a fascist have belittled it as leftist drivel. Like most political nonfiction or salacious fiction, literary journalism has not escaped the proverbial "red pen" of the censors. As such, literary journalism and political censorship can be seen as inseparable twins, especially in certain parts of the world where state-run media are still the journalistic paradigm and factual "novels" are produced to bypass censor boards.

Past research on censorship in Slovenia (Merljak Zdovc 2008), Portugal (Soares 2011; Coutinho 2020), Brazil (Lima 2011), Poland (Wiktorowska 2018; Frukacz 2019), and Argentina (Herrscher 2020) represents only a small portion of the world where literary journalists have been successful in getting stories published against the political grain, albeit disguised as travelogues or novels in some instances. Consider, for example, the case of Russia, where celebrated authors, like Turgenev in Записки охотника (*A Sportsman's Sketches*, 1852) and Tolstoy in Севастопольские рассказы (*Sevastopol Sketches*, 1855), confronted censorship with varying degrees of success, paving the way to publication for other works, such as Vladimir Gilyarovsky's Трущобные люди ([Slum people], 1887) and Vlas Doroshevich's Сахалин (*Russia's Penal Colony in the Far East*, 1902). Latin America is equally mired in the censoring of literary journalism, from Rodolfo Walsh's highly influential *Operación Masacre* (1957) and the author's open letter to the Argentine military junta to the Argentine poet/singer María Elena Walsh's op-ed "Misadventures in the Kindergarten-Country" (1993).

Unfortunately, censorship is still an important issue in the media, a trend which is on the rise due to the recent successes of political populism in once democratically determined nations. Journalistic integrity is only as stable as the political regime it scrutinizes, and literary journalism, once a salvo to combat censorship, will once again find itself as the only means to keep the world's political affronts in check.

### 2.7. Indigenous Voices

Indigenous literary journalism is the least known form of the genre and thus the least developed, but perhaps presents the most fertile research field for the years ahead. One of the most immediate concerns involves writing about the calamities facing indigenous peoples around the globe. Land appropriation is often the most salient topic, with water and mineral rights often at the heart of the struggle. More often than not, these issues camouflage the genocidal motivations behind opportunist governments' and mining companies' efforts to exploit the underprivileged and marginalized, whom they believe few people are willing to protect and defend. In Brazil, for instance, we find champions of the indigenous peoples' rights from Euclides da Cunha's dispatches for *O Estado de S.Paulo* that would later fill his book *Os Sertões* (*Rebellion in the Backlands*, 1902) to Artur Domosławski's *Śmierć w Amazonii* ([Death in Amazonia], 2013), developed from his reportage for *Gazeta Wyborcza* (2005), which investigates, among other issues, the killings of ecologists defending the rights of the inhabitants of the Praialta-Piranheira agro-extractive village in Nova Ipixuna, Brazil. A forthcoming book edited by Pablo Calvi, *The Journalist as Naturalist*, looks to expand on this direction of literary journalism a defender of indigenous rights.

While indigenous peoples around the world have a long history of documentary art, the issue raised specifically by literary journalism is one of colonialist discourse, hybrid identity, and linguistic imperialism. Like many postcolonial issues facing native novelists or playwrights of former colonies, literary journalists today in once-occupied nations need to find a balance between their native tropes and traditions and those imported by the media and professional directives of the invading culture. *Literary Journalism Studies* (spring 2018) dedicated a special issue to indigenous literary journalism, with topics covering the native peoples of Australia, the U.S., and Canada as both subject and author. Inclusivity

and hybridity are again the watchwords for the protection and expansion of indigenous literary journalism.

There are undoubtedly several other indigenous writers in once-colonized nations who have yet to be given serious academic treatment. More work, for instance, could be carried out on Brazilian, Argentine, and Peruvian indigenous literary journalism, such as works from Joseph Zárate, Natalia Viana, and Sebastián Hacher, which look into the effects of the mining industry in Peru, the high suicide rate among Brazil's indigenous peoples, and the historical abuses suffered by the Mapuche communities in Argentina. Additional studies on the Mayan author José Natividad Ic Xec and his *crónicas viajeras* (travel chronicles) would be beneficial. By examining what it means to be a literary journalist for indigenous writers, future research could explore the limits of national identity for a marginalized people, as well as their role as stakeholders in a world literary journalism.

### 2.8. Literary Journalists and (Inter)National Dailies and Magazines

While various versions of literary journalism from the nineteenth century found homes in the columns of daily papers, the majority moved toward periodicals and magazines in the twentieth century, before becoming books. But, in its collectivity, the national daily contains everything that is already in one literary journalistic article: the objective-to-subjective spectrum (lead stories to editorials); the narrative voice (the column and sports pages); and the literary influence (lifestyle and arts sections). As the national dailies gradually disappear, and the international dailies become homogenized chatter that reproduces monolithic voices of the several media conglomerates that own and operate them, literary journalism remains a case of David facing Goliath. The struggle is different in each country, of course, and research has long characterized what distinguishes literary journalistic writing in *The New Yorker* (U.S.), *XXI* (France), *piauí* (Brazil), *El Faro* (El Salvador), *Gazeta Wyborcza* (Poland), *Gatopardo* (Mexico), *Granta* (U.K.), and *Drum* or *Chimurenga* (South Africa), to name but a few. The print media are not developing (or collapsing) at the same pace in each country, and this avenue could examine the evolving relationship literary journalism has with the national and international dailies and periodicals of late.

### 2.9. Literary Journalism in the Digital Age

Mark Bowden's month-long series "Black Hawk Down" (1997) in *The Philadelphia Inquirer* (and the book published two years later) and John Branch's Pulitzer-winning article "Snow Fall: The Avalanche at Tunnel Creek" (2012) in *The New York Times* were game-changers not just in online reporting but specifically online longform narrative journalism. Yoking immersive research with multimedia literary journalism, both stories provided the sights, sounds, and even imagined smells of what it was like to dodge enemy tracers in an urban jungle or to be buried alive under a ton of snow. Amy Wilentz said of "Snow Fall" in her IALJS keynote in Paris (2014), entitled "The Role of the Literary Journalist in the Digital Era",

> So the Internet can keep us honest by letting the voices of our subjects into the conversation. But it is also demanding, and it demands, above all, action and narrative in long-form writing, because "clicks" and "eyeballs" are attracted to what is fastest moving and most cinematic in writing; clicks and eyeballs are also attracted to links and illustration, to video and photographic attachments running alongside your literary nonfiction . . . So the variety and complication of Internet presentation of nonfiction, while it may beef up a story's appearance, also can easily sully and detract from literary quality.[24]

In short, the internet offers literary journalism incredible promise and equally indelible compromise. Be careful what you wish for.

With the paper press struggling to find its place in the digital age of news reporting, literary journalism, which was often excluded from the dailies because of its excessive word count, has found new homes in the limitless html ether of sites such as Longform.com. In their article "The Digital Animation of Literary Journalism", Susan Jacobson, Jacqueline

Marino, and Robert E. Gutsche, Jr. ask whether the multimedia reportages such as these represented a passing fad or, as they conclude, a "new wave of literary journalism".[25] Examples of this global digital literary journalism abound: "Visualising the Hong Kong Protests" (*Reuters*); "How the Virus Got Out" (*The New York Times*); "Explorer, Navigator, Coloniser: Revisit Captain Cook's Legacy with the Click of a Mouse" (*Cook250*); "Living in the Unknown" (*Al Jazeera*); "The Mueller Report Illustrated" (*The Washington Post*); and "Iraq Without Water" (*Un Ponte Per*)—just to name a few.

While the internet is gradually arriving in countries once cut off from connectivity, the playing fields are not yet level. With 5G arriving in some parts of the world, other parts are just coming to terms with LAN. Research into literary journalism in the digital age remains one of the most sought-after and cutting-edge fields, destined to change as rapidly as technology itself. David Dowling, one of the leading voices in the field, has written in "Literary Journalism in the Digital Age" that such articles "induce the reader's empathy through a more immersive, integrated design, marking a distinct advance" in the field.[26] Future articles in this ninth avenue could examine the role the internet has played and likely will play in the future of countries as different in their digital consumer cultures as they are in their digital infrastructures.

## 3. Final Observations

If literary journalism is news you can read twice, world literary journalism often requires a third or even fourth pass. The reason for this, as this review article attempts to show, is fairly straight forward: the literary journalism most familiar to us is a pleasure to pick up again and again, whereas a foreign version of the genre often demands more from us as readers because of its reliance upon names, places, politics, historical references, aesthetic traditions, and cultural allusions that might, at first read, be unfamiliar. And then there is that potential bugbear: the language barrier.

Future research into international or transnational literary journalism should allow readers to walk away with new names, new titles, and new ideas linked to literary journalism and scholarly practices around the world. And, beyond the nine avenues discussed above, we are surely in need of scholarship that discusses literary journalism as a current business model, its efforts in combatting environmental skepticism and ignorance, and its role within the LGBT+ community's struggle for individual and collective freedoms and rights. Call it a buffet, a thali, tapas, a feijoada, a meza or even a smorgasbord of literary journalism, this research has as its principal goal to appeal to a multitude of tastes and appetites.

In spite of this inclusive diet, however, scholars will also discover that a majority of the literary journalism articles discussed today still hail from Europe and Latin America. The reason for this is less geopolitical than representational: save the U.S., the majority of literary journalism produced in the past century has come from these two regions. As noted earlier, fewer practitioners, for reasons which future research may be inclined to explore, and perhaps fewer literary journalism scholars are at present engaged in its writing and study. For example, Africa is experiencing a renaissance of sorts in literary journalism at present, at least in the former Anglophone and Lusophone colonies; and while that has translated into a glaring paucity of literary journalism scholarship on pan-Africa at the moment, it should provide plenty of fodder for future research.

In a similar vein, Asia remains largely under-represented. It is difficult to locate academic scholars working on literary journalism in this expansive part of the world. Perhaps there are many practitioners of literary journalism, as there are scholars who study the form, and we were just not looking in the right places or are tapping into the wrong academic disciplines. Surely, given the importance of literary journalism throughout Oceania, one would expect to find a more active presence today in the Anglo-influenced city-states of the Asia-Pacific and the countries of the subcontinent. As with African literary journalism, though, that will remain a question for future research to address.

Journalism is about information, literature about story; literary journalism is about information as story and story through information. Literary journalism studies, as this review article hopes to show its readers, is about all of this—and much, much more.

**Funding:** This research received no external funding.

**Institutional Review Board Statement:** Not applicable.

**Informed Consent Statement:** Not applicable.

**Data Availability Statement:** Not applicable.

**Conflicts of Interest:** The author declares no conflict of interest.

## Notes

1. This review article first appeared, in a slightly edited version, as the Introduction to *The Routledge Companion to World Literary Journalism*, published in December 2022.

2. For more on these three phases of postcolonialism, see Ashcroft et al. 2002, *The Empire Writes Back*. (Ashcroft et al. 2002)

3. See the "About Us" page on the IALJS's website (https://ialjs.org/about-us/), accessed on 30 April 2023.

4. The Sapir–Whorf hypothesis, which argues that one's language and culture influence that person's perception of reality, is of particular interest to the study of world literary journalisms, whose claim to authenticity and facticity are nonetheless based on linguistic renderings of truth and observation.

5. Consider as well that today's international "barista" serves coffee, while the original Italian *barista* tends the bar.

6. Leite Maia, "Alumbrar-se", pp. 371–88.

7. Warnes, "Magical Realism and the Legacy of German Idealism", p. 489.

8. Fludernik and Ryan, Introduction, p. 6. Scholarly research into the nature and limits of factual objectivity is vast, with Fludernik being one of the main references. For further reading, see, for instance, Valentin Vološinov, *Marxism and Philosophy of Language*; Tristram Hunt, "Whose Truth? Objective Truth and a Challenge for History"; Michael Kagan, "Is Truth in the Eye of the Beholder—Objective Credibility Assessment in Refugee Status Determination"; and Rafael Paes Henriques, "O problema da objetividade jornalística: duas perspectivas".

9. Tobias Eberwein, "Reconstruction of a Scandal: The Relotius Case in Germany", p. 149.

10. Fludernik and Ryan, p. 7.

11. Fludernik and Ryan, p. 1.

12. Genette, "Fictional Narrative, Factual Narrative", p. 771.

13. Genette, p. 772.

14. See note 13 above

15. Berlin and Kay, *Basic Color Terms*.

16. Ong, *Orality and Literacy*, p. 52.

17. Boas, Introduction, pp. 25–26. Boas is the oft-cited source for the myth that the Inuit people have over fifty different words for snow, depending not only their semantics (the various types and densities of snow) but also on their grammatical declinations. For more on the "hoax" of the Inuit peoples' dozens of words for snow, perpetuated by years of academic oral tradition, see Martin, "Eskimo Words for Snow": pp. 418–23.

18. Wilson and Donnan, *A Companion to Border Studies*, p. 11.

19. Wilson and Donnan, p. 1.

20. Wilson and Donnan, p. 2.

21. See note 20 above

22. Roberts, "Firing the Canon", p. 83.

23. Roggenkamp, "Journalistic Literature", pp. 81–82.

24. Wilentz, "The Role of the Literary Journalist in the Digital Era", pp. 39–40.

25. Jacobson, Marino, and Gutsche, Jr., "The Digital Animation of Literary Journalism", p. 2.

26. Dowling, "Literary Journalism in the Digital Age", pp. 530–31.

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
