# Peer review of "Notes on the Present and Future Research on World Literary Journalisms"

_journalmedia, doi:10.3390/journalmedia4030063_

Round 1

Reviewer 1 Report

This is an extremely comprehensive and substantive discussion of the current state of the field of study concerned with “literary journalism” (a term that describes non-fiction reporting that makes use of literary and fictional techniques, and also includes reportage, narrative non-fiction, longform journalism and other terms). The author/s begins by examining the various ways in which writing generally considered to fall into the scholarly category of literary journalism can use the fictional and the factual to produce a genre, and how certain types of literary journalism have emerged out of certain national, cultural and language traditions. He identifies the various terms by which these variations of the genre are known across the world. Through this exercise, he makes the case that literary journalism is a world form, with its origins in various global traditions, rather than an American invention that proliferated across borders. This is a necessary corrective to a misconception that emerged as a result of discussions around the 1960s New Journalism in American public culture, and North American scholarship on anglophone literary journalism, which arguably kicked off the field of study the writer is describing. Another important line of argument is that an attempt to pin down origins and influences is wrongheaded, as the context in which these forms emerge is so multifarious and historically complex that these factors cannot be definitively established. Instead, the writer proposes nine areas that could prove productive for further study in this growing field. These are likely to be very helpful to scholars. The article is both well grounded in relevant studies and expansive in its grasp of global literary journalism and scholarship. It should be a necessary grounding text for students and scholars in the area. However, I do suggest some minor revisions. The article is taken from the introduction to a book and references to the chapters of the book are still in it. These should be changed to reference the published book, unless this journal issue is carrying all the book chapters too, in which case, the mentions of the book and chapters should be tweaked to show that. Secondly, there are some typos in the article, so a second read for those would be good.

Language is fine

Author Response

An accurate assessment of the review article's main points/arguments, as well as two of its glaring weakness that I will remedy in the revision stage ("mentions of the book and chapters should be tweaked to show that. Secondly, there are some typos in the article, so a second read for those would be good").

Reviewer 2 Report

Overall, the article attempts to offer an overview of literary journalism’s diversity around the world and concludes with different areas of research to study literary journalism in the future. In doing so, this article becomes close to an essay, as it ambitiously moves from a worldwide view of literary journalism that continuously acknowledges its diversity, while at the same time attempting to put it into a single voice across borders.

In the end, the article seems to offer a good outline of worldwide study of literary journalism, but sadly does not offer specific examples of literary journalism. Nor does it give a good definition for literary journalism, thus assuming the reader knows it by heart. In this sense, this article would be enriched with a small footnote where it delves briefly on a possible definition for literary journalism. A mention of IALJS’s perspective featured in its website, for example, or something of sorts, should be enough to solve this.

Some moments also require some rethinking and rewriting: Line 82-86 states that readers have a knowledge of its potential limits and drawbacks, but I am not sure most academics – the self-proclaimed experts on the subject – would even agree on said limits and drawbacks. In line 42-45 the author states that literary journalism is like a religion, with many mutations catalyzed by periods of great human confluence; it’s a bold statement and I believe it’s an exaggeration for dramatic purposes. In 593-596 the author states that future research should give readers new names, new titles and new ideas; and while I agree, I wish the author would have done this for those that practice literary journalism and not just for those that study it. Indeed, the author knows and mentions worldwide scholars of literary journalism, but there is a lack of specific examples of literary journalism (which appear only briefly in the section War and Conflict).

Possible typos: "Nowhere on planet coffee is one substituted for the other." & "We do not, for instance, refer to reportages as faction and novels as nonfaction."

Author Response

I thank the reviewer for his or her comments and suggestions, which I have duly integrated.

Per the critique: "sadly does not offer specific examples of literary journalism," I have put in titles of world literary journalism where I felt they were needed. Given the nature of the review article, none of these examples was meant to be discussed in detail, let alone analyzed -- just pointed out.

Per the suggestion: "Nor does it give a good definition for literary journalism, thus assuming the reader knows it by heart." Any reader picking up the book that this piece introduced (or any reader coming to this separate article) would already have his or her own definition of literary journalism in mind (per their culture or nation). I did add a brief mention of IALJS's definition to use a touchstone, if only to upend it later in the discussion.

Per the query about the sentence: "Most readers drawn to this article will already have a working knowledge of what literary journalism is or means to them, what it represents today for the future of print and digital media, and what its potential limits and drawbacks are." It is true that, for the book, a reader drawn to this specific collection would be already well versed in at least their nation's version of literary journalism. One would not just arrive at this book by chance. On the other hand, a reader of this separate article might not know a lot of the problems and drawbacks, as the reviewer rightly points out.  I have thus added the sentence: "For those readers who not possess such prior knowledge of the genre, nor of its current global praxes, this article will present nine of what I consider to be the most pressing issues facing literary journalism in the world today, and, by extension, literary journalism studies."

Per the suggestion: "In line 42-45 the author states that literary journalism is like a religion, with many mutations catalyzed by periods of great human confluence; it’s a bold statement and I believe it’s an exaggeration for dramatic purposes."  Sorry, but I believe the extended metaphor is apt, in particular because religion is one of the main reasons by the form has developed in various forms around the world. I thus kept it intact.

Per the comment: "those that practice literary journalism and not just for those that study it." Agreed, but my field is not the writing but rather the study of literary journalism. Philology and creating writing are distinct disciplines. I am not a literary journalist by trade, nor have ever taught the practice of literary journalism. In a similar vein, I am not a poet and thus would not weigh in how one should teach the writing of poetry.

Per the comment: "lack of specific examples of literary journalism". Again, I have peppered titles around the piece, but the idea behind this introduction is precisely not to take the thunder away from those authors who do pen chapters on specific examples. There were all excised for reasons of transitioning this piece from an introduction to their work to a stand-alone piece.

Reviewer 3 Report

The paper offers a review on the current state of world literary journalism researches and current trends of research, suggesting nine areas of research the author(s) considers as under-represented currently. The paper is well-organized and very cogent in its argument, featuring a thorough bibliography that aims to cover worldwide research on literary journalism. There are two aspects that still have room for improvement, though. The first one are statements regarding the differences in LJ across countries and cultures (e.g. lines 279 and 342): short examples that show HOW those traditions of LJ are different would further strenghten the argument. The second point is the criticism of objective truth. While the paper relies on more tangible aspects such as the influence of language in the perception of reality (and the discussion on color could be expanded to include other aspects of it, such as the Sapir-Whorf hypothesis), current scholarly criticism of factual objectivity would certainly be welcome (eg. Valentin Voloshinov in "Marxism and Philosophy of Language", Tristram Hunt in "Whose Truth? Objective Truth and a Challenge for History", Michael Kagan in  "Is Truth in the Eye of the Beholder - Objective Credibility Assessment in Refugee Status Determination", and Rafael Paes Henrique in "O problema da objetividade jornalística: duas perspectivas"). 

Author Response

I thank the reviewer here for suggesting new titles that explore factual objectivity, and I will place them in a footnote. Most important among them was the reference to the S-W hypothesis and linguistic determinism, which I have worked briefly into the discussion. Thank you for that.

Per the comment: “short examples that show HOW those traditions of LJ are different would further strengthen the argument”. A fair point, but these traditions are precisely discussed in each of the 45 chapters that comprise the book for which this piece served as an introduction. I fully understand the need to include some, now that this is a stand-alone piece, and I will make a brief mention here in two of the spots identified, but examining in-depth the “How” is rather complex and clearly beyond the scope of this review article.

Reviewer 4 Report

The subject of this paper is very important and relevant: to propose future lines for the practice and study of Literary Journalism. It is also written in a creative way.

The following positive aspects should be highlighted: i) This article addresses a relevant issue on literary journalism; ii) The author masters the main issues of the field of study; iii) The author makes a complete mapping of the challenges of the discipline; iv) The author presents an original - and to some extent creative - approach to the issues.

However, the paper has two weaknesses: i) it does not make explicit the methodology used, ii) nor does it refer precisely to the concepts. 

The following improvements are suggested:

i)               Explanation of the methodology that allowed the author to arrive at the proposals he describes;

 The author should define what he understands by Literary Journalism, as the concept covers a range of diverse objects and phenomena. The concept of Literary Journalism has to be defined. Although it is said (p. 2) that the LJ does not need definition, I understand that the author, under the cover of this concept, refers to a vast set of texts, genres, phenomena, which are not LJ. Right in the introduction, the author says that the LJ is timeless and ubiquitous, which seems to be symptomatic of a confusion between narrative and literary journalism. Still in this part of the article, he considers the LJ a 'genre' that can take many forms and styles and, at the same time a 'format': it seems to me, however, that it is abusive to consider that everything can be LJ as long as it has factually based narrative format and stylistic work. In short, it is necessary to review the concept, to define it (indeed authors like Keeble, cited in the references, can make a relevant contribution to this).

The author suggests that the term should be plural (p. 1), but he/she does not provide grounds for this.

The author does not explain why he believes he should develop what he calls ‘nine avenues’ - nine paths for future research in the field of studies (p. 6). Moreover, since he does not distinguish LJ as a genre or format from LJ as a discipline, it is not clear who these nine avenues serve; the LJ or the study of LJ?

ii)             An academic paper should avoid using excessively metaphorical and connotative language, even when its subject is the literary journalism. The author uses this register a lot.

 From a formal point of view, it is suggested:

1.        The abstract should mainly explain which are the nine areas to be developed in the future of the field of Literary Journalism (LJ).

2.        The keywords should be more complete, touching on aspects of the proposals made in the article.         

Extensive editing of english language required.

Author Response

I appreciate the reviewer’s “positive aspects” noted in the article, but I do disagree strongly on many of the “negative” aspects pointed out (e.g., that there is no “methodology” here, per se, to describe or discuss LJ further—given the nature of the piece as a book introduction precisely, defining world literary journalism was to be done through its various chapters. To address this concern, I have since added a brief definition of LJ), above all the quality of the English in this piece.

I am fully against (perhaps this is an English and/or American issue) stolid and lifeless academic prose and have intentionally written this introduction in a lively, if at times unorthodox or irreverent, register. I stand by that decision. Were I, perhaps, a junior colleague whose name was new to the field, I would no doubt have sided against using such a tone throughout; but I am not, and will not temper my English here to fit the presumed “standard” voice of academic writing. Should this be a serious problem for the journal editors, I will willingly remove my piece for consideration from publication.